# T2K Latest Results on Neutrino–Nucleus Cross-Sections †

Andrew Cudd ⬤ on behalf of the T2K Collaboration

University of Colorado Boulder, Boulder, CO 80309, USA; andrew.cudd@colorado.edu
† Presented at the 23rd International Workshop on Neutrinos from Accelerators, Salt Lake City, UT, USA, 30–31 July 2022.

**Abstract:** A detailed understanding of neutrino–nucleus interactions is essential for the precise measurement of neutrino oscillations at long baseline experiments, such as T2K. The T2K near detector complex, designed to constrain the T2K flux and cross-section models, also provides a complementary program of neutrino interaction cross-section measurements. Through the use of multiple target materials (carbon, water, lead, iron), and the ability to sample different neutrino spectra (with detectors located on- and off-axis with respect to the beam direction), T2K is able to investigate atomic number and energy dependence of interaction cross-sections in a single experiment. In particular, T2K has recently performed the first joint on-/off-axis measurement of the charged current channel without pion in the final state. Furthermore, dedicated efforts are devoted to investigating rare or poorly studied interaction channels. Indeed, an improved analysis of the coherent pion production cross-section was recently accomplished, including an anti-neutrino sample for the first time. Those results, together with an overview of the T2K measurement strategy, adopted to reduce the model dependence, will be presented in these proceedings.

**Keywords:** neutrino; cross-section; T2K

## 1. The T2K Experiment

The Tokai to Kamioka (T2K) experiment [1] is a long-baseline neutrino oscillation experiment designed to provide measurements of neutrino oscillation parameters through muon (anti-)neutrino disappearance and electron (anti-)neutrino appearance. T2K consists of a relatively pure (anti-)muon neutrino beam produced at the J-PARC facility that travels approximately 295 km across Japan to the Super-Kamiokande (SK) detector and through a series of near detectors positioned 280 m from the target. T2K utilizes an off-axis neutrino beam where SK is placed 2.5 degrees away from the center of the neutrino beam, which produces a narrower neutrino energy spectrum peaked at 0.6 GeV. In order to accurately measure the oscillation probability, precise knowledge of the underlying neutrino interaction is required. A recent T2K measurement [2] shows that the cross-section error is currently one of the leading sources of systematic uncertainty.

T2K utilizes a suite of detectors placed 280 m downstream from the target to monitor the neutrino beam and measure the unoscillated neutrino spectra to provide constraints for the oscillation analysis and perform neutrino interaction cross-section measurements. INGRID [3] is the on-axis detector and is primarily used to measure the neutrino beam position and profile. INGRID is built from fourteen identical iron and plastic scintillator modules arranged in a cross shape centered on the beam axis and extending to about 1.1 degrees off-axis. A special module with only plastic scintillator bars called the Proton Module was placed on axis for several T2K data runs and provides a higher resolution tracking detector and target for neutrino cross-section measurements.

ND280 [4] is the off-axis detector at 2.5 degrees in line with SK and comprises several sub-detectors inside the refurbished UA1/NOMAD magnet, which provides a 0.2 T magnetic field. The inner tracking region is composed of the $\pi^0$ detector, two fine-grained detectors (FGDs) as a neutrino target with tracking capabilities, and three gaseous argon

time projection chambers (TPCs) for charged particle tracking and identification. FGD1 only contains plastic scintillator bars whereas FGD2 contains scintillator bars and passive water layers for a water target. The tracking region is surrounded by an electromagnetic calorimeter (ECAL) for measuring photons and track/shower separation and plastic scintillator bars instrumented in the magnet yoke called the side muon range detector (SMRD) to act as a veto for particles entering from outside the detector.

## 2. Cross-Section Analysis Strategy

The analyses presented in these proceedings use a binned maximum likelihood method to perform a fit to the number of selected events, minimizing the following test statistic:

$$\chi^2 = \chi^2_{\text{stat}} + \chi^2_{\text{syst}} = -2\ln\mathcal{L}_{\text{stat}} - 2\ln\mathcal{L}_{\text{syst}}. \tag{1}$$

The statistical likelihood is the Poisson log-likelihood ratio with modifications based on the Barlow–Beeston method [5] to include the effect of finite Monte Carlo statistics. The unfolding is performed by including a set of normalization parameters for each truth kinematic bin that are allowed to freely vary the signal cross-section, known as the template likelihood method. Prior uncertainties for the systematic (or nuisance) parameters are included in the likelihood as a Gaussian penalty term. Included in each analysis are systematic parameters for the neutrino flux model, the detector model, and the neutrino interaction model (primarily for background events). The result is the best-fit number of selected signal events and a covariance matrix describing the correlations between each of the fit parameters, which are used to extract the cross-section.

The flux-integrated cross-section is extracted to avoid model dependence from assumptions on the shape of the neutrino energy spectrum. The differential flux-integrated cross-section is defined as:

$$\frac{\mathrm{d}\sigma}{\mathrm{d}x_i} = \frac{\hat{N}_i^{\text{signal}}}{\epsilon_i^{MC}\Phi N_{\text{nucleons}}^{FV}} \times \frac{1}{\Delta x_i}, \tag{2}$$

where $x_i$ is the variable(s) used for the cross-section extraction, $\hat{N}_i^{\text{signal}}$ is the number of selected signal events in a given bin $i$ as determined from the fit, $\epsilon_i^{MC}$ is the efficiency in each bin, $\Phi$ is the integrated neutrino flux, $N_{\text{nucleons}}^{FV}$ is the number of nucleons in the fiducial volume, and $\Delta x_i$ is the bin width. As indicated in Equation (2), the event rate is normalized by the total integrated flux instead of correcting for the flux in each bin.

## 3. On-/Off-Axis Analysis

This analysis is a joint fit to both on-axis (INGRID) and off-axis (ND280) data extracting the double differential charged-current zero pion (CC0$\pi$) cross-section on hydrocarbon as a function of the outgoing muon kinematics (momentum and angle, where the angle is with respect to the neutrino beam direction) [6]. The main benefits of the joint analysis of on-axis and off-axis data are a reduction in flux uncertainties from the correlations between the positions, and to study the energy dependence of the cross-section in a single analysis. Signal events are CC0$\pi$ interactions, defined by a neutrino interaction with an outgoing muon, zero pions in the final state, and any number of final-state hadrons. CC0$\pi$ was chosen as a signal because it is the most common event type in the T2K oscillation analysis. The signal selection for ND280 requires the interaction vertex to be within the FGD1 fiducial volume (to select interactions only on hydrocarbon), and after passing the selection cuts to identify the muon, any protons, and reject pions, the signal events are separated into five signal samples as follows:

- Sample I ($\mu$TPC): defined by a single muon candidate in the TPCs and no other particles;
- Sample II ($\mu$TPC + pTPC): a muon candidate in the TPCs with one or more proton candidates in the TPC;

- Sample III (μTPC + pFGD): a muon candidate in the TPCs and a proton candidate in FGD1;
- Sample IV (μFGD + pTPC): a muon candidate in FGD1 (possibly reaching the ECAL) and a proton candidate in the TPC;
- Sample V (μFGD): a muon candidate in FGD1 (possibly reaching the ECAL) and no other particles.

The ND280 selection also includes three control samples in the analysis to constrain the background event rate from events producing one or more pions. Similarly, the signal selection for INGRID requires the interaction vertex to be within the Proton Module fiducial volume and requires events to have exactly one muon-like track and zero or one proton-like track. The INGRID selection has only one signal sample and one control sample (for single-pion events) in the analysis.

The signal and control samples from both ND280 and INGRID are included in the likelihood fit along with a set of systematic parameters and a covariance matrix describing the priors as described in the previous section. A selected subset of angular bins of the extracted cross-section as a function of muon momentum is shown in Figure 1. The binning of the result was chosen to cover as much muon kinematic phase space as possible while removing areas with no efficiency for measuring events.

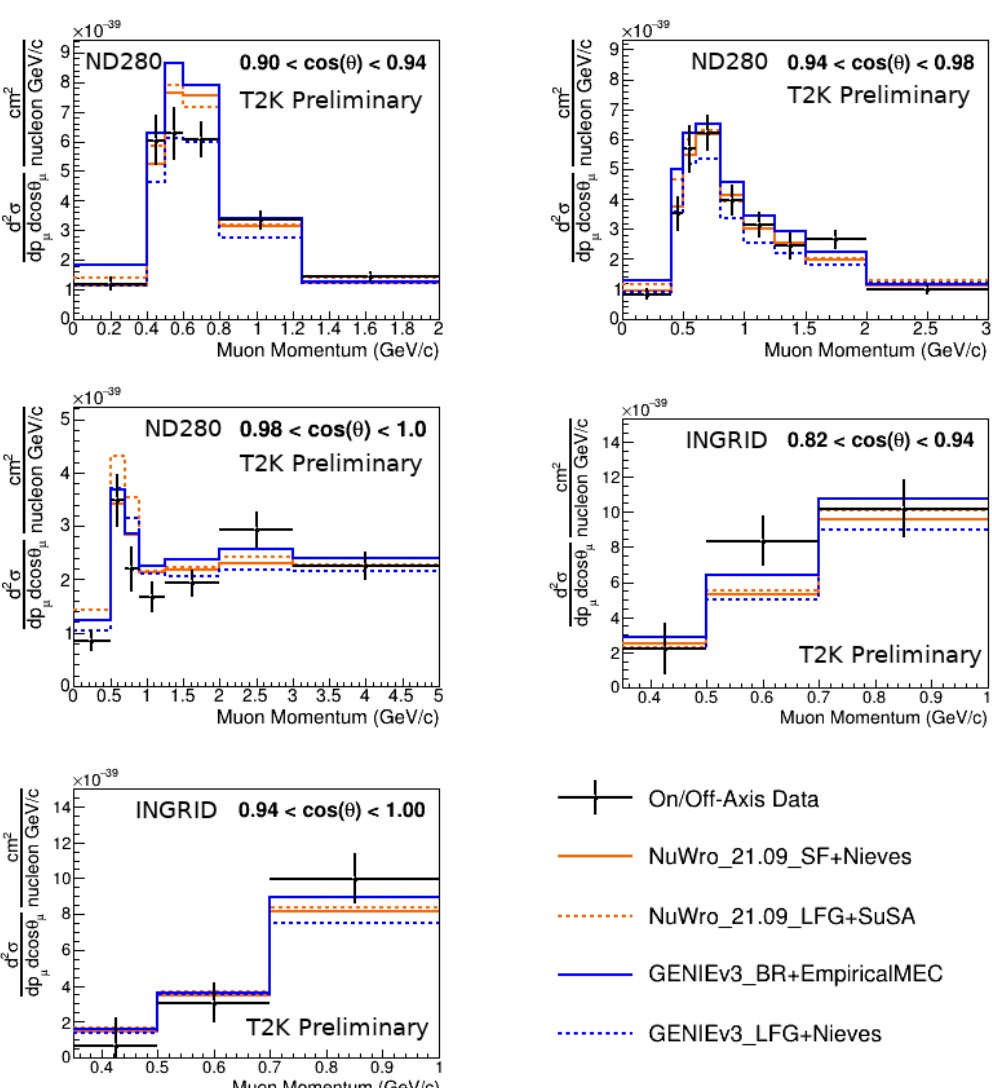

**Figure 1.** ND280 and INGRID measured cross-sections in selected forward angle bins as a function of muon momentum with several model comparisons.

Overall, the Monte Carlo model comparisons do not describe the data well as measured by the $\chi^2$ per degree of freedom (*d*). The full measurement has 70 kinematic bins giving a $\chi^2/d \sim 2$, with the $\chi^2$ for each model in Table 1. Generator models continue to struggle to describe the data, and more work is needed for both model and generator development along with more precise and novel measurements.

**Table 1.** Agreement between this result and the various model comparisons as measured by the $\chi^2$ for both the joint result and when compared to each detector individually. ND280 has 58 cross-section bins and INGRID has 12 cross-section bins for a combined 70 total bins.

| Model | ND280 | INGRID | Joint |
|---|---|---|---|
| NuWro SF + Nieves | 122.74 | 15.68 | 137.02 |
| NuWro LFG + SuSAv2 | 121.57 | 11.13 | 135.38 |
| GENIE BRRFG + EmpMEC | 141.40 | 12.80 | 156.05 |
| GENIE LFG + Nieves | 125.50 | 14.45 | 135.69 |

## 4. Coherent Pion Production Analysis

This analysis is a total cross-section measurement of charged current coherent pion production for both muon neutrino and anti-neutrino modes. Neutral current coherent pion production is an important background for the oscillation analysis because the only signature of the interaction is a produced $\pi^0$ decaying to two photons and can be mistaken for an electron-like appearance signal. By measuring the charged-current version of the process, constraints can be placed on the neutral current process. Additionally, neutrino-induced coherent pion production has historically not been a well-understood process.

Signal events are events with a negative (positive) muon and positive (negative) charged pion for neutrino (anti-neutrino) mode where the interaction vertex was in the FGD1 fiducial volume (to select interactions only on hydrocarbon). Cuts are placed to select events with low momentum transfer squared ($|t| < 0.15$ GeV$^2$) and low energy deposited around the vertex (VA < 15 MeV) (also referred to as vertex activity). These are designed to remove events with low-energy untracked particles that deposit energy near the vertex. As shown in Figure 2, coherent signal events are concentrated at low $|t|$ and low vertex activity. This analysis also includes two control regions for events that produce additional pions and for single pion events with high $|t|$ that comprise the main background.

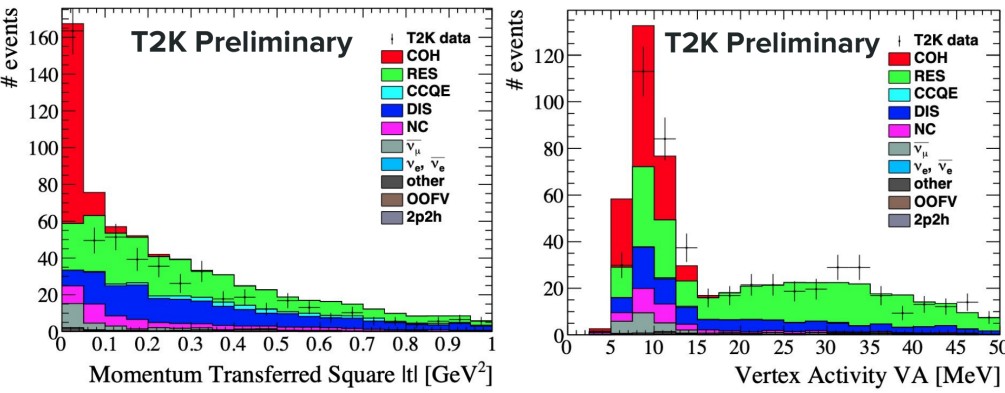

**Figure 2.** Charged current coherent pion selection for neutrino mode as a function of momentum transfer squared and vertex activity with data overlaid. Note that the signal events are concentrated at low $|t|$ and low vertex activity.

The flux-integrated cross-section on carbon is extracted using the likelihood fit as described in a previous section. The total cross-section for FGD1 is assumed to be a sum across each individual element present in the target material with some nuclear-scaling factor. The cross-section for carbon can then be calculated with knowledge of the material composition of the target and choosing a nuclear-scaling factor, whereas, for this analysis,

the carbon cross-section was measured using nuclear-scaling factors of $F(A) = A^{1/3}$ and $F(A) = A^2$. The $A^{1/3}$ comes from the paper by Rein and Sehgal [7], describing the coherent pion production cross-section and the $A^2$ factor comes from the fact that the interaction is coherent across all nuclei in the nucleus. The measured cross-sections for neutrinos and anti-neutrinos using the $A^{1/3}$ factor to match the literature are shown in Figure 3. The measured cross-sections for both neutrino and anti-neutrino match fairly close to the more recent Berger–Sehgal model [8] compared to the original Rein–Sehgal model [7]. The total cross-section and uncertainty for each scaling factor are:

$$\sigma_{^{12}C}^{\text{CC-COH}} = \begin{cases} 2.98 \pm 0.37 \,(\text{stat.}) \pm 0.58 \,(\text{syst.}) \times 10^{-40} \,\text{cm}^2 & \left(F(A) = A^{1/3}\right) \\ 2.69 \pm 0.33 \,(\text{stat.}) \pm 0.52 \,(\text{syst.}) \times 10^{-40} \,\text{cm}^2 & \left(F(A) = A^2\right) \end{cases}. \quad (3)$$

The difference between the total cross-section when using either nuclear-scaling factor is comparable or smaller than the total uncertainty, making it difficult to draw a conclusion on which factor is more accurate.

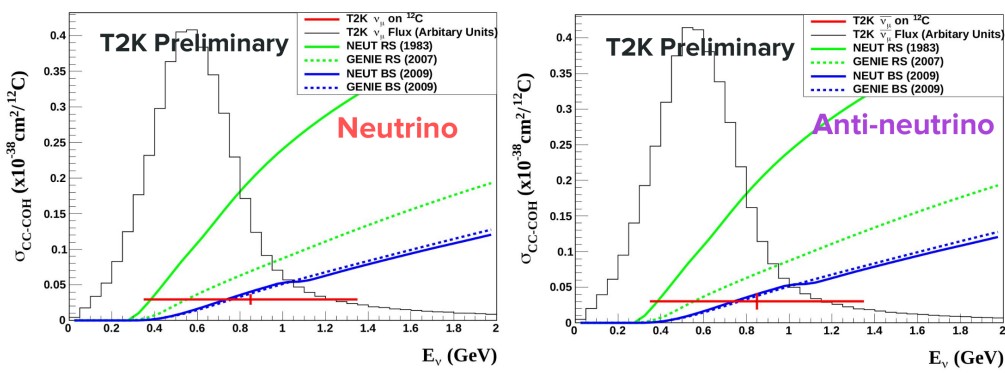

**Figure 3.** Charged current coherent pion total cross-section on carbon using $A^{1/3}$ scaling for neutrino (**left**) and anti-neutrino (**right**) compared to several Monte Carlo predictions using the Rein–Sehgal (RS) and Berger–Sehgal (BS) models.

**Funding:** This research received no external funding.

**Institutional Review Board Statement:** Not applicable.

**Informed Consent Statement:** Not applicable.

**Data Availability Statement:** Not applicable.

**Conflicts of Interest:** The author declares no conflict of interest.

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
