# Peer review of "T2K Latest Results on Neutrino–Nucleus Cross-Sections†"

_psf, doi:10.3390/psf8010014_

Round 1

Reviewer 1 Report

Thank you for these very interesting and informative proceedings. The summary is very well written. Hereby you can see only a few minor suggestion: While the detector is very well explained, it is unclear what is FGD1 (vs. FGD2 or 3) better to just write FGD or explain the difference / location and why FGD1 was the one to be chosen. Figure 1, Given the on and off axis, it is not entirely clear how the angular bins are defined? Theta is the scattering angle from which axis?  Figure 2, It will be worth mentioning the values of the cuts on t and vertex activity, as you’ve already shown the plots.  Figure 3, it is unclear which factor was used to get the results in the figures.  I suggest adding a small conclusions section (or even two additional sentences). In addition to the comparison between BS and RS models, your thoughts on the chosen factor could be mentioned. 

Thank you again for your contribution.  

Author Response

While the detector is very well explained, it is unclear what is FGD1 (vs. FGD2 or 3) better to just write FGD or explain the difference / location and why FGD1 was the one to be chosen.

- Added some text to clarify the difference between FGD1 and FGD2 along with some text to explain why only FGD1 was used.

Figure 1, Given the on and off axis, it is not entirely clear how the angular bins are defined? Theta is the scattering angle from which axis?

- Angular bins are based on the muon kinematics and detection efficiency (e.g. INGRID has little to zero efficiency for measuring backward going muons). The scattering angle is with respect to the neutrino beam direction, which has been clarified in the text.

Figure 2, It will be worth mentioning the values of the cuts on t and vertex activity, as you’ve already shown the plots.

- Cut values have been added to the text.

Figure 3, it is unclear which factor was used to get the results in the figures.  I suggest adding a small conclusions section (or even two additional sentences). In addition to the comparison between BS and RS models, your thoughts on the chosen factor could be mentioned.

- The scaling factor used is mentioned in the text, and has been added to the caption of Figure 3 to improve clarity. I have also added a comment on how the choice of scaling factor affects the result.